# Whole-Genome Sequence Analysis of *Candida glabrata* Isolates from a Patient with Persistent Fungemia and Determination of the Molecular Mechanisms of Multidrug Resistance

**DOI:** 10.3390/jof9050515

**Published:** 2023-04-26

**Authors:** Ha Jin Lim, Min Ji Choi, Seung A. Byun, Eun Jeong Won, Joo Heon Park, Yong Jun Choi, Hyun-Jung Choi, Hyun-Woo Choi, Seung-Jung Kee, Soo Hyun Kim, Myung Geun Shin, Seung Yeob Lee, Mi-Na Kim, Jong Hee Shin

**Affiliations:** 1Department of Laboratory Medicine, Chonnam National University Medical School and Chonnam National University Hospital, Gwangju 61469, Republic of Korea; hajin00905@naver.com (H.J.L.);; 2Microbiological Analysis Team, Biometrology Group, Korea Research Institute of Standards and Science (KRISS), Daejeon 34113, Republic of Korea; minji1246@naver.com; 3Department of Laboratory Medicine, Asan Medical Center, University of Ulsan College of Medicine, Seoul 05505, Republic of Korea; 4Department of Laboratory Medicine, Jeonbuk National University Medical School and Jeonbuk National University Hospital, Jeonju 54907, Republic of Korea; 5Research Institute of Clinical Medicine of Jeonbuk National University-Biomedical Research Institute of Jeonbuk National University Hospital, Jeonju 54907, Republic of Korea

**Keywords:** whole-genome sequencing, *Candida glabrata*, multidrug resistance, resistance mechanisms

## Abstract

Whole-genome sequencing (WGS) was used to determine the molecular mechanisms of multidrug resistance for 10 serial *Candida glabrata* bloodstream isolates obtained from a neutropenic patient during 82 days of amphotericin B (AMB) or echinocandin therapy. For WGS, a library was prepared and sequenced using a Nextera DNA Flex Kit (Illumina) and the MiseqDx (Illumina) instrument. All isolates harbored the same Msh2p substitution, V239L, associated with multilocus sequence type 7 and a Pdr1p substitution, L825P, that caused azole resistance. Of six isolates with increased AMB MICs (≥2 mg/L), three harboring the Erg6p A158fs mutation had AMB MICs ≥ 8 mg/L, and three harboring the Erg6p R314K, Erg3p G236D, or Erg3p F226fs mutation had AMB MICs of 2–3 mg/L. Four isolates harboring the Erg6p A158fs or R314K mutation had fluconazole MICs of 4–8 mg/L while the remaining six had fluconazole MICs ≥ 256 mg/L. Two isolates with micafungin MICs > 8 mg/L harbored Fks2p (I661_L662insF) and Fks1p (C499fs) mutations, while six isolates with micafungin MICs of 0.25–2 mg/L harbored an Fks2p K1357E substitution. Using WGS, we detected novel mechanisms of AMB and echinocandin resistance; we explored mechanisms that may explain the complex relationship between AMB and azole resistance.

## 1. Introduction

*Candida* bloodstream infections (BSIs) are the most common nosocomial fungal infections and are associated with high rates of mortality [1,2]. *Candida albicans* is the most common species causing candidemia; however, the proportion of candidemia cases caused by non-*albicans Candida* (NAC) species (e.g., *Candida glabrata*, *Candida parapsilosis*, and *Candida tropicalis*) is increasing worldwide [3]. The increasing frequency of BSI isolates of NAC species is associated with many different factors such as antifungal drug exposure, catheter use, intensive care unit admission, age, and geographic distribution [2,4]. Recent increases in antifungal use have led to increasing azole resistance among BSI isolates of NAC species; the emergence of multidrug-resistant (MDR) *Candida* strains, such as *Candida auris* and *C. glabrata*, is a serious concern [3,5]. *C. auris* is an emerging MDR yeast that has caused healthcare-associated outbreaks in numerous countries [5]. We previously reported the first three cases of *C. auris* fungemia [6]; since that report, *C. auris* isolates have rarely been obtained from blood cultures in Republic of Korea [7]. In contrast, the increasing number of BSI isolates of *C. glabrata* with antifungal resistance and the emergence of MDR strains are serious public health problems in Republic of Korea [8,9].

*C. glabrata* is a natural commensal yeast in the human gut, genitourinary tract, or oral cavity; however, it can cause BSI that have high mortality rates [10,11], and it exhibits innately low susceptibility to azole drugs, especially fluconazole. *C*. *glabrata* isolates are no longer considered susceptible to fluconazole; they are classified only as fluconazole-susceptible dose-dependent (F-SDD) or fluconazole-resistant (FR) [12]. *C. glabrata* can rapidly develop resistance during the course of antifungal therapy, probably due to its haploid nature and ability to undergo genomic changes [4,8,13,14,15]. According to the global SENTRY study, the incidence of FR *C. glabrata* isolates increased from 8.6% to 10.1% during 1997–2014 and the incidence of echinocandin-resistant *C. glabrata* isolates during 2006–2016 was 1.7% to 3.5% [3]. Acquired azole resistance in *C. glabrata* is most commonly mediated by overexpression of the drug-efflux transporter genes *CgCDR1*, *CgCDR2*, and *CgSNQ2* through a gain-of-function (GOF) mutation in the transcription factor pleiotropic drug resistance (*PDR*)*1* [8]; other mechanisms may also contribute. The mutations typically responsible for echinocandin resistance are *FKS1* alterations in the most prevalent *Candida* species. Among *C. glabrata* isolates, point mutations of *FKS1* and *FKS2* are the most common resistance mechanisms. *FKS1* and *FKS2* encode 1,3-β-D-glucan synthase, the target of echinocandins; therefore, mutations in *FKS* are associated with resistance to anidulafungin, caspofungin, and micafungin in *C. glabrata* [16,17]. More importantly, 5.5% to 7.6% of FR *C. glabrata* isolates are co-resistant to echinocandins and are thus considered MDR [3]. Although amphotericin B (AMB) resistance is still uncommon among *C. glabrata* isolates, MDR clinical isolates of *C. glabrata* are being increasingly identified and have presented significant management challenges in recent years [4,9,18,19].

*C. glabrata* has emerged as one of the most common causes of invasive infections in specific subsets of patients, including hematopoietic stem cell transplant recipients, who are commonly placed on prophylactic antifungal regimens [20,21]. We previously reported the emergence of MDR *C. glabrata* BSI isolates from a neutropenic patient who had undergone hematopoietic stem cell transplantation due to acute myeloid leukemia [9]. Multiple *C. glabrata* BSI isolates that showed various resistance patterns to azoles, echinocandins, and AMB were recovered over alternating therapeutic courses of echinocandin and AMB. All 10 isolates showed sequence type (ST) 7, as revealed by multilocus sequence typing (MLST), and had indistinguishable karyotypes [9]. All isolates exhibited high-level (two isolates had micafungin minimum inhibitory concentrations (MICs) > 8 mg/L) or low-level (eight isolates had micafungin MICs of 0.12–0.5 mg/L) echinocandin resistance. Six isolates exhibited AMB resistance (MIC ≥ 2 mg/L by the ETEST^®^). Interestingly, four F-SDD isolates exhibited AMB resistance, while four FR isolates exhibited AMB susceptibility. Although we found a novel insertion in the hotspot (HS) region in *FKS2* in two isolates with high-level echinocandin resistance by targeted sequencing of the region, more information is needed to explain the molecular mechanisms of multiple non-sequential resistance profiles involving azoles, echinocandins, and AMB for these serial isolates. Whole-genome sequencing (WGS) has been utilized to elucidate mechanisms of drug resistance in *Candida* species [22,23]. Thus, WGS of *C. glabrata* may detect several mutations in different genes involved in the ergosterol biosynthesis pathway (e.g., *ERG6* and *ERG3*; AMB resistance) or in those that encode transcription factors that regulate efflux pump expression (e.g., *PDR1*; azole resistance), together with *FKS1/2* mutations (echinocandin resistance). Therefore, we performed WGS of the same 10 serial MDR *C. glabrata* BSI isolates to investigate the mechanism of AMB resistance, the mechanism underlying the inverse relationship between AMB and azole resistance, the mechanism responsible for low-level echinocandin resistance, and the evolutionary process based on the molecular mechanisms present in this clonal population of *C. glabrata*.

## 2. Materials and Methods

### 2.1. Fungal Isolates and Antifungal Susceptibility Testing

All 10 serial BSI isolates of *C. glabrata* from our previous report were assessed [9]. The patient was treated with micafungin for 23 days from hospital day (HD) 82 to 104, with AMB for 16 days from HD 105 to 120 and for 40 days from HD 124 to 163, and with caspofungin for 6 days from HD 121 to 126 and for 16 days from HD 147 to 162. The 10 *C. glabrata* isolates tested in this study were recovered serially from blood cultures between HDs 99 and 160 [9]. All isolates were stored at −70 °C in trypticase soy broth supplemented with 15% glycerol. The antifungal MICs of fluconazole, voriconazole, posaconazole, itraconazole, anidulafungin, caspofungin, and micafungin were re-determined using the Sensititre YeastOne^®^ system (Thermo Fisher Scientific, Waltham, MA, USA) whereas the antifungal MICs of AMB were determined using ETEST^®^ (bioMérieux, Marcy-l’Étoile, France). Two reference strains, *Candida parapsilosis* ATCC 22019 and *Candida krusei* ATCC 6258, were included in each antifungal susceptibility test as quality control isolates. The MIC interpretive criteria included species-specific Clinical and Laboratory Standards Institute (CLSI) clinical breakpoints for fluconazole (resistant, ≥64 mg/L; susceptible dose-dependent, ≤32 mg/L), anidulafungin (resistant, ≥0.5 mg/L; intermediate, 0.25 mg/L), caspofungin (resistant, ≥0.5 mg/L; intermediate, 0.25 mg/L), and micafungin (resistant, ≥0.25 mg/L; intermediate, 0.12 mg/L) [12]. The epidemiological cutoff values (ECVs) proposed in CLSI M59-ED3 or European Committee on Antimicrobial Susceptibility Testing (EUCAST; AMB only) were used as MIC interpretive criteria for AMB (susceptible, ≤1 mg/L; resistant, >1 mg/L), voriconazole (susceptible, ≤0.25 mg/L; resistant, >0.25 mg/L), posaconazole (susceptible, ≤1 mg/L; resistant, >1 mg/L), and itraconazole (susceptible, ≤4 mg/L; resistant, >4 mg/L) [24,25]. In this study, MDR was defined as resistance to two or more classes (triazoles/echinocandins/polyenes) of antifungal drugs [18]. Therapeutic failure was defined as either persistence of *Candida* in the bloodstream despite >72 h of antifungal therapy or the development of breakthrough fungemia during treatment with the indicated antifungal agents for >72 h [8,26].

### 2.2. Whole-Genome Sequencing

DNA was extracted from 10 *C. glabrata* serial isolates (isolates 1–10) as described previously [27]. A library was prepared using a Nextera DNA Flex Kit (Illumina, San Diego, CA, USA) and sequenced as 150-bp paired-ends using the MiseqDx (Illumina) instrument. The sequencing matrix was extracted using Sequencing Analysis Viewer version 2.4.7 (Illumina). Adapter sequences and low-quality bases were trimmed using BBDuk from BBMap package version 38.95 [28]. The trimmed sequences were aligned to the reference genome using Burrows-Wheeler Aligner version 0.7.17 with the BWA-MEM algorithm [29]. The *C. glabrata* reference genome (CBS 138) from the *Candida* Genome Database was used as the reference genome [30]. Duplicate marking and conversion to the BAM file were performed using Picard version 2.27.4 [31]. Using HaplotypeCaller in GATK version 4.2.6.1, variants including single-nucleotide polymorphisms (SNPs) and insertions and deletions (INDELs) were called. Variants were filtered as described previously [32] and those that had a depth below 10 were removed. Annotation was carried out using snpEff version 4.3t with GFF file version 3 from the *Candida* Genome Database [30]. To filter the variants further, a list of genes associated with antifungal resistance was curated using the file CBS138 of chromosomal features from the *Candida* Genome Database [30] with the keywords ‘(drug) resistance’, ‘resistant’, ‘antifungal (target)’, and ‘(suppress/reduced) sensitivity’. *ERG* genes, *FCY1*, *FCY2*, *FKS3*, *HSP90* (*HSC82*), *MMR1* (*CAGL0A04169g*), *NDT80*, *TAC1* (*HAL9*), and *UPC2* (*UPC2A* and *UPC2B*), which were missing from the initial keyword search but needed for further investigation of three classes of antifungal agents and MDR, were added to the list [33,34,35]. Thus, as the criteria for shortlisting MDR genes, we used 182 genes associated with antifungal resistance (Appendix A). These included the representative *ERG* genes, *CDR1*, *CDR2* (*PDH1*), *FEN1*, *FKS1*, *FKS2*, *FKS3*, *FLR1*, *SNQ2*, *PDR1*, and *QDR2* [36]. Additionally, phylogenetic analysis was performed based on SNP data for 182 resistance genes, using the maximum-likelihood method with the Kimura two-parameter model and bootstrap analysis with 1000 replications in MEGA version 11.0.11 [37]. Nonsynonymous mutations associated with antifungal resistance and correlated with the antifungal MICs were visually inspected using Integrative Genomics Viewer version 2.14.0 (Appendix A). This study was approved by the Ethics Committee of Chonnam National University Hospital (CNUH) Gwangju, Korea; the need for informed parental consent was waived due to the retrospective nature of the study (CNUH-2014-290).

### 2.3. In Vivo Virulence Analysis Using Galleria Mellonella

We evaluated the virulence of five serial isolates (isolates 1–5) of *C. glabrata* and 35 BSI isolates of *C. glabrata* obtained from Korean multicenter surveillance cultures (18 FR isolates harboring Pdr1p mutations, and 17 F-SDD isolates without Pdr1p mutations) [8] in the *G. mellonella* insect model, as described previously [38,39]. Briefly, groups of 20 larvae (~150 mg; S-worm, Cheonan, Republic of Korea) were stored in wood shavings in the dark at 18 °C prior to use. The following three control groups were included: larvae injected with 10 µL of phosphate-buffered saline (N = 20), larvae that received needle injury only (N = 20), and untouched larvae (N = 20). A Hamilton syringe (25 gauge, 50 µL) was used to inoculate larvae with *C*. *glabrata*; it was also used to apply treatment or control solutions to the larvae. To determine the virulence of clinical *C. glabrata* isolates, larvae were infected with 5 × 10^6^ conidia per larvae; survival was monitored up to 96 h post-infection at 37 °C. Data were combined to calculate the mean percentage survival.

### 2.4. Statistical Analysis

RStudio version 2022.7.1.554 (RStudio, Inc., Boston, MA, USA) was used for statistical analysis. The Wilcoxon rank-sum test or Student’s *t*-test was used to determine the significance of between-group differences in survival at 24, 48, 72, and 96 h, based on the Shapiro–Wilk normality test and F-test. Differences were considered statistically significant at *p <* 0.05.

### 2.5. Deposition of the Raw Sequence Data

The raw sequence data were deposited in the NCBI Sequence Read Archive (BioProject PRJNA949257).

## 3. Results

In the WGS analysis, each run matrix was within the manufacturer’s recommended value (Appendix A) [40]. An average of 5,222,511 reads were produced per isolate and 98.6% of the total reads were mapped to the reference genome (CBS138) with 55.4× to 70.8× coverage (average 61.6×). After variant calling, a total of 90,650 mutations (9601 INDELs and 81,049 SNPs) were detected per isolate; 12.8% were nonsynonymous mutations (Table 1). When filtered according to the resistance genes in which nonsynonymous mutations were detected, an average of 251 mutations (16 INDELs and 235 SNPs) were observed per isolate, of which 238 (94.8%) were simultaneously observed in all isolates. We detected 0.421 to 0.438 SNPs/kb among 10 isolates. Phylogenetic analysis based on WGS SNP data for the 182 resistance genes showed considerable diversity among 10 isolates, regardless of isolation date or antifungal susceptibility pattern (Appendix A).

Nonsynonymous mutations in various genes associated with antifungal resistance were compared with the antifungal MICs of the 10 serial isolates (isolates 1–10). All isolates showed the same Msh2p substitution, V239L (associated with ST7 in MLST). All isolates also had the same nonsynonymous mutations in *CDR1* (H58Y), *PDH1* (E839D and T1530K), *PDR1* (S76P, V91I, L98S, T143P, and L825P), *QDR2* (T199M), *FLR1* (V254I), *FKS3* (I3T, A42G, K206E, N865S, R1472Q, and F1768I), and *FEN1* (M155T). Figure 1 presents the major mutations in the *ERG* genes (*ERG1–ERG10*), *FKS1/2*, and *MSH2* and *PDR1* GOF mutations detected by WGS. Of the *ERG* genes, 11 unique nonsynonymous mutations were detected in *ERG2*, *ERG3*, *ERG4*, *ERG6*, *ERG7*, *ERG8*, and *ERG10* throughout the serial isolates, of which 6 SNPs (*ERG2* I207V, *ERG4* T13N, *ERG6* R48K, *ERG7* T732A, *ERG8* N448S, and *ERG10* N107D) were observed in all 10 isolates. When the AMB MICs were investigated, three isolates (isolates 3, 6, and 7) harboring a frameshift mutation (A158fs) in *ERG6* showed strong resistance to AMB (MICs of 8–16 mg/L). The other three isolates showing increased MICs against AMB (2–3 mg/L) harbored *ERG6* R314K (isolate 5), *ERG3* G236D and *ERG4* P227fs (isolate 8), and *ERG3* F226fs (isolate 10) mutations.

With regard to azole resistance, a *PDR1* GOF mutation (L825P) was observed in the 10 serial isolates. Six FR isolates (isolate 1, 2, 4, and 8–10) showed markedly high MICs (≥256 mg/L) for fluconazole and higher MICs for voriconazole, posaconazole, and itraconazole. However, the fluconazole MICs were 4–8 mg/L in four (isolate 3 and 5–7) isolates that harbored *ERG6* A158fs or R314K simultaneously. These four isolates also showed lower MICs for other azoles. With regard to echinocandins, all of the isolates exhibited intermediate to high resistance to at least one of anidulafungin, caspofungin, and micafungin. Of the isolates, isolate 2 and 5, which exhibited MICs > 8 mg/L for three echinocandins, harbored an *FKS2* I661_L662insF mutation in combination with *FKS1* C499fs. The isolate harboring an F659del HS mutation in *FKS2* (isolate 9) showed definitively increased anidulafungin, caspofungin, and micafungin MICs (2, >8, and 1 mg/L, respectively). In contrast, the isolate simultaneously harboring *FKS2* F659del and S201fs (isolate 4) was susceptible to anidulafungin and micafungin, and intermediate only to caspofungin. The other isolates (isolate 1, 3, 6–8, and 10) showed slightly increased echinocandin MICs (at least two of three echinocandins, ≥0.25 mg/L) and harbored a K1357E mutation in *FKS2*.

Figure 2 depicts the possible evolution of the antifungal mechanisms of the 10 sequential clonal *C. glabrata* isolates with the same Pdr1p L825P mutation during the course of AMB or echinocandin therapy. All 10 isolates were associated with breakthrough fungemia during the administration of echinocandins (isolates 1 and 4), AMB (isolates 2, 3, and 5–8), or both (isolates 9 and 10). The Fks2p K1357E mutation first appeared after 17 days of micafungin exposure in isolate 1 and it was shared by five subsequent isolates (isolates 3, 6, 7, 8, and 10). These six isolates were designated as subpopulation #1. The Erg6p A158fs mutation first appeared after 16 days of AMB therapy in isolate 3 (subpopulation #1–2) and was shared by two subsequent isolates (isolate 6 and 7; #1–2). Clonal subpopulation #1 re-appeared after 6 days of caspofungin therapy and 35 days of AMB therapy with the addition of Erg4p P227fs combined with an Erg3p G236D mutation (isolate 8; subpopulation #1–3), and after 19 days of caspofungin and 52 days of AMB therapy with an additional Erg3p F226fs mutation (isolate 10; subpopulation #1–4). Overall, echinocandin breakthrough fungemia was caused by two isolates (isolate 1 and 10) of subpopulation #1, which harbored the Fks2p K1357E mutation. On the other hand, Fks2p I661_L662insF and Fks1p C499fs appeared in isolate 2 after 23 days of micafungin therapy; these mutations were shared by isolate 5, so isolates 2 and 5 were designated as clonal subpopulation #2. Clonal subpopulation #2 with an additional Erg6p R314K mutation appeared after 22 days of AMB therapy (isolate 5; subpopulation #2–2). In addition, two isolates (isolate 4 and 9) had a unique Fks2p mutation (designated as clonal subpopulations #3 and #4, respectively).

Table 2 shows the virulence in the *G*. *mellonella* model, among *C. glabrata* isolates 1 to 5. In vivo assays in the insect *G. mellonella* revealed that the 96-h survival rates of *G. mellonella* larvae infected with four isolates (isolates 1, 2, 3, and 5) were relatively higher than survival rates of *G. mellonella* larvae infected with FR or F-SDD isolates. The mean survival rate in larvae infected with FR isolates (N = 18) was significantly higher than the rate in larvae infected with F-SDD isolates (N = 17) at all four time periods examined (24 h, *P* = 0.010; 48 h, *P* = 0.003; 72 h, *P* = 0.002; 96 h, *P* = 0.006).

## 4. Discussion

The development of resistance to *C. glabrata* BSI isolates during treatment is a possible cause of treatment failure, but few reports have provided a comprehensive understanding of how *C. glabrata* genomes can accumulate gene mutations that result in phenotypic resistance to antifungals during an extended course of antifungal therapy. In the present study, all 10 isolates harbored the same Msh2p substitution, V239L, which is known to be associated with both MLST type ST7 [8] and hypermutability [41]. All isolates harbored the same Pdr1p L825P mutation, which is associated with azole resistance [8]. Our WGS showed that the 10 isolates had a relatively low density of SNPs (0.421–0.438 SNPs/kb), reflecting their clonal nature, and that the genetic changes in antifungal drug-associated genes were due to long-term antifungal therapy [36,42]. An important implication of our findings is the high concordance between several nonsynonymous mutations in genes affecting AMB or echinocandin resistance and their MICs. For the first time, we have demonstrated that the presence of Erg6p mutations in *C. glabrata* isolates with Pdr1p GOF mutations could lower fluconazole MICs. 

Acquired AMB resistance in *Candida* isolates is rare [43,44,45,46]. The rare occurrence of AMB resistance in *C. glabrata* may be partly due to a lack of detection ability using current CLSI or the European Committee on Antimicrobial Susceptibility Testing reference methods. In this study, the AMB MICs of six isolates were ≥2 mg/L by the ETEST^®^, but those of all 10 isolates were 0.5–1 and 0.5–2 mg/L by the CLSI M27 method and Sensititre Yeast One^®^ system, respectively (data not shown), in agreement with a previous report [47]. The limited studies available suggested a mechanistic role for *ERG2*, *ERG3*, *ERG4*, and *ERG6* in AMB resistance [44,45,46,48,49]. Previous studies reported a nonsense mutation [44] and missense mutation [50] in *ERG6* that resulted in AMB resistance due to a composition change in sterol, which is the target of polyene. Here, we showed that three isolates of *C. glabrata* harboring a disruptive frameshift mutation (A158fs) in *ERG6* exhibited markedly increased MICs (8–12 mg/L) for AMB and that harboring a substitution mutation (R314K) in *ERG6* moderately increased the AMB MIC to 2 mg/L. Thus, *ERG6* may be involved in AMB resistance in *C. glabrata*. An I207V mutation in *ERG2* was also detected but was found in all isolates (isolates 1–10). Mutations in *ERG3* or *ERG4* have been found in AMB-resistant *Candida albicans* [48,51,52] and *Saccharomyces cerevisiae* [53], but rarely in *C. glabrata* [54]. In the present study, two isolates with AMB MICs of 2–3 mg/L harbored *ERG3* G236D and *ERG4* P227fs (isolate 8) and *ERG3* F226fs (isolate 10) mutations, which may require more supporting evidence. 

In our previous study, by comparing the *PDR1* sequences of each *C*. *glabrata* isolate with the reference *PDR1* sequence of *C. glabrata* (GenBank accession no. FJ550269) [55], we demonstrated that nearly all FR BSI isolates of *C. glabrata* in Korea harbored FR-specific Pdr1p mutations by excluding MLST genotype-specific Pdr1p amino acid substitutions [8]. In this study, all 10 isolates had an FR-specific nonsynonymous mutation (L825P) in *PDR1*, which may mediate azole resistance in *C. glabrata* [8]. However, among the isolates, four (isolate 3 and 5–7) showed low azole MICs (F-SDD) despite a *PDR1* GOF mutation (L825P), while six had a fluconazole MIC ≥ 256 mg/L (FR). All four F-SDD isolates harbored an Erg6p (A158fs or R314K) mutation. A previous study showed that the lower ergosterol content associated with a nonsense mutation in *ERG6* may have an indirect effect on susceptibility to azoles by preventing the targeting of efflux pumps to the plasma membrane, thereby favoring the accumulation of these drugs within the cell [44]. The presence of *ERG6* mutations could lead to defects in ergosterol synthesis and changes in the binding of the efflux pump. In our previous study, FR isolates of *C. glabrata* exhibited higher mean expression levels of *CgCDR1*, *CgCDR2*, and *CgSNQ2*, compared with F-SDD isolates [8]. When we compared the expression levels of *CgCDR1*, *CgCDR2*, and *CgSNQ2* in five isolates with the same Pdr1p L825P mutation, without (isolates 1, 2, and 4; FR isolates) or with (isolates 3 and 5; F-SDD isolates) an Erg6p mutation, the expression levels of *CgCDR1* and *CgSNQ2* in the three FR isolates were relatively higher than levels in the two F-SDD isolates, and similar to the mean expression levels of *CgCDR1* and *CgSNQ2* in 30 FR isolates harboring Pdr1p mutations. Taken together, our findings indicate that *C. glabrata* isolates with the same Pdr1p GOF mutations do not always show the same FR result—they can be F-SDD in AMB-resistant isolates with Erg6p mutations.

Although sequencing of the HS regions in *FKS* genes is the most convenient way of determining echinocandin resistance mechanisms, mutations occurring outside of these HS regions can also lead to echinocandin therapeutic failure, which confirms the importance of sequencing the entire *FKS* gene [4,56]. In the present WGS analysis, four isolates (isolate 2, 4, 5, and 9) showed disruptive INDELs in HS regions of *FKS2*, and all isolates (except isolate 9) showed missense or frameshift mutations occurring outside of these HS regions. Of the isolates, two with anidulafungin, caspofungin, and micafungin MICs > 8 mg/L harbored not only the mutation Fks2p I661_L662insF but also Fks1p C499fs. Given that mutations in *FKS1* or *FKS2* [26,57], or the combination of a null function mutation in *FKS1* and point HS mutation in *FKS2* [56], could lead to strong resistance among *C. glabrata* strains, the unique HS mutations (I661_L662insF and F659del) in *FKS2* found in this study may have different impacts on echinocandin MICs according to the combination of other *FKS* nonsynonymous mutations. There were also two isolates harboring the mutation *FKS2* F659del with or without the upstream *FKS2* S201fs mutation. Relatively strong resistance to echinocandins was observed in an isolate harboring a single F659del mutation in *FKS2* (isolate 9), but the echinocandin MIC was slightly decreased in an isolate harboring both mutations (F659del and S201fs) in *FKS2* (isolate 4). The reasons for the lowered echinocandin MIC in isolate 4 harboring the F659del and S201fs mutations are uncertain. One possibility is that the upstream *FKS2* S201fs mutation may affect the downstream *FKS2* F659del mutation, but more evidence is needed. Six isolates in subpopulation #1 harboring the Fks2p K1357E mutation (isolates 1, 3, 6–8, and 10) showed micafungin MICs of 0.25–2 mg/L. 

The Fks2p K1357E mutation first appeared as breakthrough fungemia after 17 days of micafungin exposure in isolate 1, and five additional isolates were recovered from HD 99 to HD 160 despite further micafungin (5 days) or caspofungin (19 days) therapy. The role of Fks2p K1357E in echinocandin resistance remains uncertain as this SNP has not been described previously. However, a previous report showed that *C. glabrata* isolates harboring a single non-HS mutation in an *FKS* gene showed slightly increased MICs for echinocandins [58]. Here, we found that the echinocandin breakthrough fungemia was caused by two isolates (isolate 1 and 10), suggesting that the Fks2p K1357E mutation is associated with echinocandin therapeutic failure. Overall, our WGS study suggests that isolates harboring nonsynonymous mutations located outside the HS regions in *FKS* genes can increase echinocandin resistance. 

*C. glabrata* BSI isolates from a particular geographic area have been reported to comprise a small number of major STs, according to MLST analysis. MLST of Korean BSI isolates showed that ST7 (47.8%) was the most common type, followed by ST3 (22.5%); the remaining isolates exhibited 28 types of minor STs [59]. FR isolates of *C. glabrata* typically had one Pdr1p amino acid substitution, which were rarely shared by two isolates from the same hospital in the same year, in agreement with a previous report that *C. glabrata* transmission between patients is rare [60]. Although our isolates exhibited ST7, the most common ST in Republic of Korea, none collected in 2009–2018 harbored Pdr1p L825P mutation except our 10 isolates, suggesting independent development of FR in *C. glabrata* in most patients [8,60]. Despite the clonal nature of the BSI isolates of *C. glabrata* obtained from our patient, the serial isolates showed significant non-serial phenotypic MIC variations to AMB, azole, or echinocandins. Similarly, phylogenetic analysis by WGS showed substantial genetic diversity, regardless of isolation date and phenotypic antifungal susceptibility pattern (Appendix A).

We postulated that subpopulations with different resistance profiles are likely to have persisted in the gut and alternately invaded the bloodstream under selective pressure, highlighting the adaptability of *C. glabrata* to long-term treatment with various antifungal agents [9]. In the present study, WGS enabled us to detect the possible molecular mechanism responsible for the low- and high-level antifungal resistance of each isolate of *C. glabrata* and to show the evolution of molecular mechanisms within the same subpopulation due to different resistance profiles. Our findings suggest that some nonsynonymous mutations found in the same subpopulation (subpopulations #1 and #2) may represent pre-existing mutations, and some new mutations occurred after antifungal drug exposure. Subpopulations with pre-existing mutations are likely to persist in the gut or other mucosal sites and appear in the bloodstream with or without new genetic changes during the long course of antifungal therapy.

The fitness cost related to antifungal resistance acquisition by *C. glabrata* is unclear, and few studies have been reported thus far [61,62,63]. *G. mellonella* has been used as a host model to study *C. glabrata* virulence and antifungal efficacy [64]. In the present study, the mean survival rate in larvae infected with FR isolates was significantly higher than the rate in larvae infected with F-SDD isolates at all four time periods examined, indicating that F-SDD isolates without *PDR1* mutations may be more virulent than FR isolates harboring *PDR1* mutations. Moreover, our results suggest that our MDR *C. glabrata* isolates with *PDR1* gene mutations have reduced virulence in the *G. mellonella* model.

A notable limitation of this study is that, although many of the detected mutations were located in genes involved in resistance, we did not directly assess their roles in resistance. The reintroduction of mutant alleles into susceptible strains via gene editing would be a useful approach for determining their roles in resistance. We used WGS to detect specific genetic alterations associated with antifungal resistance in serial clonal *C. glabrata* isolates. Some of these newly detected mutations were out of the target region in the gene (e.g., non-HS regions in *FKS1/2*) or out of the target gene (e.g., *ERG* genes) from our previous study based on conventional sequencing [9]. In this study, novel HS mutations (F659del mutations) were detected by WGS in isolates 4 and 9.

## 5. Conclusions

In conclusion, this study provides important perspectives on the utility of WGS for detecting molecular mechanisms of multidrug resistance based on 10 serial *C. glabrata* BSI isolates obtained from a patient with breakthrough fungemia during extended AMB or echinocandin therapy. Pdr1p GOF and Fksp mutations in *C. glabrata* may not always have the same effects; they may cause different levels of antifungal resistance, depending on the combination of nonsynonymous mutations present. Fluconazole MICs are lower in *C. glabrata* isolates with the same Pdr1p GOF mutation than in AMB-resistant isolates with Egr6p mutations. Fks2p HS mutations combined with Fks1p null-function mutations contribute to high-level echinocandin resistance. In addition, Fks2p mutations outside HS regions contribute to low-level echinocandin resistance. Persistent subpopulations of *C. glabrata* undergoing continuous clonal genetic evolution during long-term antifungal therapy could be responsible for the non-serial multiple antifungal resistance phenotypes of *C. glabrata* BSI isolates. In conclusion, WGS will improve the detection and monitoring of molecular mechanisms of antifungal resistance.

## Figures and Tables

**Figure 1 jof-09-00515-f001:**
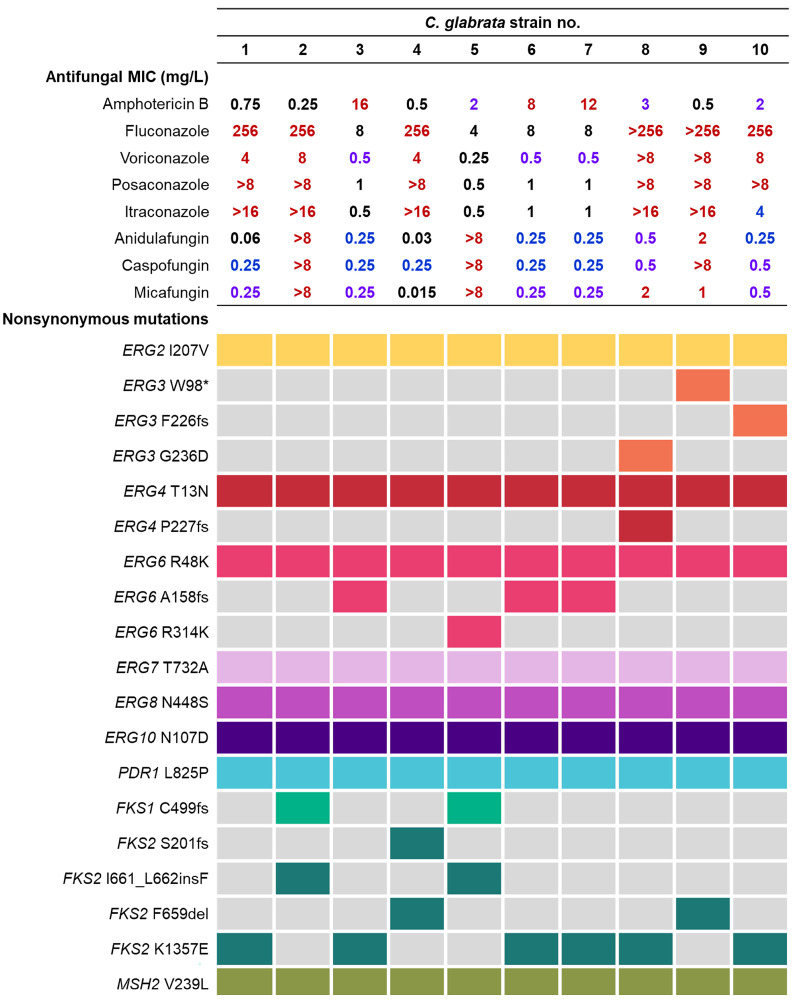
Results of antifungal susceptibility testing and whole-genome sequencing of 10 bloodstream isolates of *C. glabrata* isolated serially from a patient. Antifungal MICs were interpreted according to the clinical breakpoints or epidemiologic cut-offs of the Clinical and Laboratory Standards Institute guidelines (M60-ED2 and M59-ED3, respectively), and the categories are highlighted with colors (red: highly resistant; purple: resistant; blue: not resistant but intermediate, susceptible-dose dependent, or showing an increased MIC).

**Figure 2 jof-09-00515-f002:**
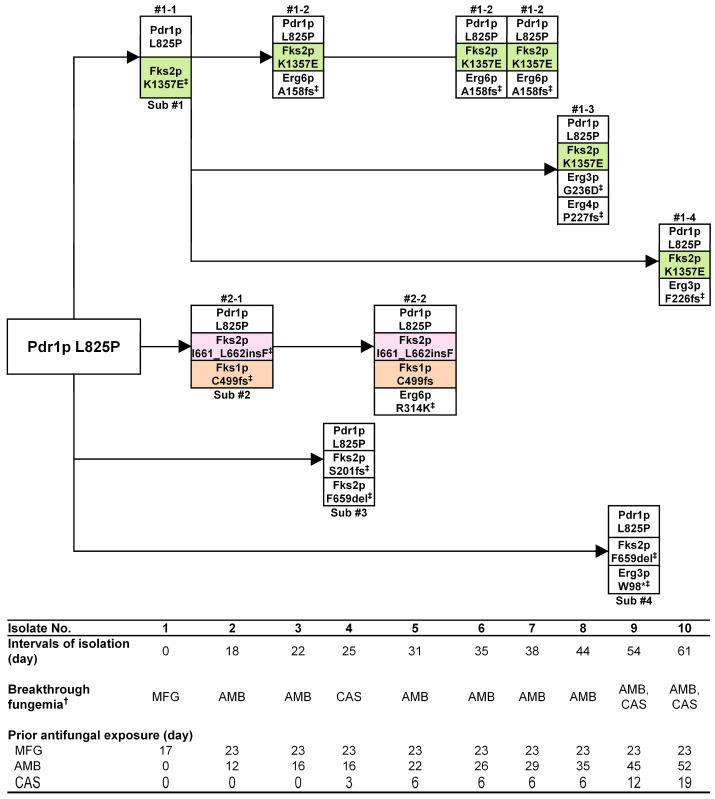
Possible evolution of the antifungal resistance mechanisms of the *C. glabrata* isolates with the Pdr1p L825P mutation during the course of echinocandin and amphotericin B therapy, as determined by whole-genome sequencing. Persisting mutations in each subpopulation were colored green, pink, or orange. ^†^ Breakthrough fungemia was defined as fungemia that developed during treatment with the indicated antifungal agents for >72 h. ^‡^ Newly appearing mutations are marked in each subpopulation. Abbreviations: MFG, micafungin; AMB, amphotericin B; CAS, caspofungin; sub, subpopulation.

**Table 1 jof-09-00515-t001:** Mutations detected in the 10 serial isolates using whole-genome sequencing *.

Isolate No.	Total Mutation			Nonsynonymous Mutation			Nonsynonymous Mutation in the Resistant Gene ^†^		
	INDEL	SNP	Total	INDEL	SNP	Total	INDEL	SNP	Total
1	9377	79,839	89,216	849	10,669	11,518	15	234	249
2	9646	81,239	90,885	872	10,814	11,686	18	229	247
3	9657	81,113	90,770	840	10,771	11,611	14	235	249
4	9578	81,220	90,798	869	10,820	11,689	17	233	250
5	9624	81,107	90,731	834	10,749	11,583	16	235	251
6	9643	81,252	90,895	847	10,833	11,680	17	236	253
7	9590	81,029	90,619	829	10,797	11,626	16	239	255
8	9653	81,374	90,027	839	10,800	11,639	14	236	250
9	9607	81,004	90,651	844	10,773	11,617	17	235	252
10	9633	81,268	90,901	843	10,812	11,655	17	235	252
Total	96,008	810,485	906,493	8466	107,838	116,304	161	2347	2508
Average	9601	81,049	90,650	847	10,784	11,630	16	235	251
SD	83	439	519	14	48	53	1	3	2

Abbreviations: SD, standard deviation; INDEL, insertion, and deletion; SNP, single nucleotide. * The *C. glabrata* reference genome (CBS 138) was used. ^†^ The resistance genes are listed in Appendix A.

**Table 2 jof-09-00515-t002:** In vivo virulence of five serial *C. glabrata* isolates in the *G. mellonella* model, compared with blood isolates from Republic of Korean multicenter surveillance cultures.

Isolate No.	Antifungal Susceptibility *			Survival Rate (%) of Infected *G. mellonella*			
	FLU	AMB	MFG	24 h	48 h	72 h	96 h
Serial isolates in this study							
1	R	S	I	80.0	80.0	80.0	80.0
2	R	S	R	90.0	82.5	75.0	62.5
3	SDD	R	I	70.0	60.0	60.0	55.0
4	R	S	S	100.0	80.0	70.0	35.0
5	SDD	R	R	97.5	90.0	80.0	77.5
Blood isolates from Korean multicenter surveillance cultures (Mean ± SD)							
FR (N=18)	R	S	S	93.7 ± 14.2 ^†^	82.6 ± 24.3 ^†^	64.7 ± 25.7 ^†^	49.6 ± 24.0 ^†^
F-SDD (N=17)	R	S	S	83.8 ± 6.8	61.5 ± 14.0	38.4 ± 21.2	25.7 ± 25.6

* Interpretative categories of antifungal resistance determined using the Clinical and Laboratory Standards Institute (CLSI) CLSI M60-ED [12] or EUCAST ECOFF (AMB only) [25]. ^†^
*P* < 0.05 between FR and F-SDD groups. Abbreviations: SD, standard deviation; FR, fluconazole-resistant; F-SDD, fluconazole-susceptible dose-dependent; FLU, fluconazole; AMB, amphotericin B; MFG, micafungin; R, resistant; SDD, susceptible dose-dependent; S, susceptible; I, intermediate

## Data Availability

The raw sequencing data were deposited in the NCBI Sequence Read Archive (bioproject PRJNA949257).

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
