# Peer review of "Whole-Genome Sequence Analysis of Candida glabrata Isolates from a Patient with Persistent Fungemia and Determination of the Molecular Mechanisms of Multidrug Resistance"

_jof, 2023, doi:10.3390/jof9050515_

Round 1

Reviewer 1 Report

This study by Lim et al explains some novel mechanisms of AMB and echinocandin resistance, and mechanisms explaining the complicated relationship between AMB and azole resistance. The authors have performed whole genome sequencing of blood stream isolates of C. glabrata from a patient with persistence fungemia. Although the manuscript is nicely written and explains about novel mechanisms of multi-drug resistance in C. glabrata, some minor comments can be addressed: 

1. There are various emerging species of Candida such as C. auris, and C. tropicalis. Why does this study only deal with C. glabrata?

2. Abstract section contains information regarding instruments and methodology. The Abstract is missing key information regarding the impact of this research. 

3. Introduction: Line 41, Authors have mentioned common Candida species. But they have not specified about any common Candida species. Introduction starts with C. glabrata without citing any background information regarding incidence of systemic candidiasis, risk factors, and evolution of Candida species.

4. Introduction: Line 67, authors have mentioned about gene FKS2 without giving any basic information about the function of this gene. Similarly, most genes (e.g. PDR1) are introduced directly, without their full form or their basic function, which may not be understood by common readers. 

5. Line 128, What was the criteria for shortlisting multi-drug resistance genes?

6. In Figure1 legend, change term Candida glabrata to C. glabrata.

7. The Conclusion section is very short. The authors can include more details on novel molecular mechanisms that have been identified in this research and relate it to previously known mechanisms. 

8. In Conclusion section, the authors can include information regarding the major contribution of this research in the advancement to the field of study.

Reviewer 2 Report

The manuscript by Lim et al., describes resistance in C. glabrata hospital isolates from one patient over time. The manuscript is well written and the data well presented. However it is a bit too simplistic and can benefit from additional information to give it more depth as follows:

1- It is of great interest to determine phylogenetic relatedness and epidemiological assessment of the isolates, with implications as far as nosocomial infection and strain microevolution. Authors can use SNP data, or MLST data to construct trees. 

2- Resistance in Candida is not only an issue of presence of mutations or not. For many genes such as CDR/MDR resistance is due to an increase in transcription levels. As such Reverse Transcription PCR or northern blots on select mutants might be very informative in determining resistance mechanisms.

3- Many of the mutations detected are in genes known to be involved in resistance but this does not prove that they do play a role in resistance. Reintroduction of mutant allele into sensitive strains would be a good approach to determine so.

4-It would be of interest to determine phenotypes of these mutants. This would give weight to the role these mutations are performing. Are they strong biofilm formers? Do these mutations upregulate ergosterol resulting in resistance? Are they more or less virulent compared to sensitive strains? etc..

Round 2

Reviewer 2 Report

Authors performed the required modifications.

Author Response

Thank you in advance for your careful consideration of our work.